# Human Coronavirus 229E Infection Inactivates Pyroptosis Executioner Gasdermin D but Ultimately Leads to Lytic Cell Death Partly Mediated by Gasdermin E

**DOI:** 10.3390/v16060898

**Published:** 2024-06-01

**Authors:** Xavier Martiáñez-Vendrell, Jonna Bloeme-ter Horst, Roy Hutchinson, Coralie Guy, Andrew G. Bowie, Marjolein Kikkert

**Affiliations:** 1Molecular Virology Laboratory, Leiden University Center of Infectious Diseases (LU-CID), Leiden University Medical Center, 2333 ZA Leiden, The Netherlands; x.vendrell@lumc.nl (X.M.-V.);; 2School of Biochemistry and Immunology, Trinity Biomedical Sciences Institute, Trinity College Dublin, D02 PN40 Dublin 2, Irelandagbowie@tcd.ie (A.G.B.)

**Keywords:** HCoV-229E, 3C-like protease, main protease, Mpro, virus-induced cell death, virus-induced pyroptosis, caspase-3, GSDMD, GSDME

## Abstract

Human coronavirus 229E (HCoV-229E) is associated with upper respiratory tract infections and generally causes mild respiratory symptoms. HCoV-229E infection can cause cell death, but the molecular pathways that lead to virus-induced cell death as well as the interplay between viral proteins and cellular cell death effectors remain poorly characterized for HCoV-229E. Studying how HCoV-229E and other common cold coronaviruses interact with and affect cell death pathways may help to understand its pathogenesis and compare it to that of highly pathogenic coronaviruses. Here, we report that the main protease (Mpro) of HCoV-229E can cleave gasdermin D (GSDMD) at two different sites (Q29 and Q193) within its active N-terminal domain to generate fragments that are now unable to cause pyroptosis, a form of lytic cell death normally executed by this protein. Despite GSDMD cleavage by HCoV-229E Mpro, we show that HCoV-229E infection still leads to lytic cell death. We demonstrate that during virus infection caspase-3 cleaves and activates gasdermin E (GSDME), another key executioner of pyroptosis. Accordingly, GSDME knockout cells show a significant decrease in lytic cell death upon virus infection. Finally, we show that HCoV-229E infection leads to increased lytic cell death levels in cells expressing a GSDMD mutant uncleavable by Mpro (GSDMD Q29A+Q193A). We conclude that GSDMD is inactivated by Mpro during HCoV-229E infection, preventing GSDMD-mediated cell death, and point to the caspase-3/GSDME axis as an important player in the execution of virus-induced cell death. In the context of similar reported findings for highly pathogenic coronaviruses, our results suggest that these mechanisms do not contribute to differences in pathogenicity among coronaviruses. Nonetheless, understanding the interactions of common cold-associated coronaviruses and their proteins with the programmed cell death machineries may lead to new clues for coronavirus control strategies.

## 1. Introduction

Coronaviruses (CoVs) are enveloped positive-sense single-stranded RNA (+ssRNA) viruses that belong to the family *Coronaviridae*, suborder *Cornidovirineae*, and order *Nidovirales* [1]. CoVs infect and cause disease in a wide range of animals, including birds, reptiles and mammals [2]. To date, seven CoVs are known to infect humans, all of them causing respiratory disease. Human coronaviruses (HCoVs) OC43, HCoV-HKU1, HCoV-229E, and HCoV-NL63 are associated with upper respiratory tract infections and mainly cause common cold symptoms [3]. In contrast, the severe acute respiratory syndrome CoV (SARS-CoV), the Middle East respiratory syndrome CoV (MERS-CoV), and the most recently identified SARS-CoV-2 can lead to severe pathology in the lower respiratory tract [4,5,6].

HCoV-229E was isolated from the nasal discharge of a young person with common cold symptoms in the 1960s and later identified as the first HCoV [7], and it was also the first HCoV ever to be cultured using standard tissue culture [8]. Differently from the highly pathogenic HCoVs, which belong to the genus *Betacoronavirus*, HCoV-229E falls under the genus *Alphacoronavirus* [3]. HCoV-229E utilizes the aminopeptidase N (APN or CD13) expressed on cellular membranes as the functional receptor, and enters the cell using the endocytic pathway [9,10]. Endosomal acidification promotes membrane fusion leading to the release of the viral genome into the cytoplasm. HCoV-229E has a genome of approximately 27.5 kb [11,12]. Upon release and uncoating of the genomic RNA in the cytoplasm, the open reading frames (ORFs) 1a and 1b at the 5′ of the genome are immediately translated into the viral replicase polyproteins pp1a and, upon a ribosomal frameshift, pp1ab. The replicase polyproteins are then co- and post-translationally processed by the viral proteases (the papain-like proteases PLP1 and PLP2, and the main protease) contained within pp1a and pp1ab in order to release the 16 non-structural proteins (nsps) that will assemble the viral replication machinery [13].

As for many nsps, the coronavirus proteases have been described to be polyfunctional proteins, meaning they possess accessory functions other than their main role in the proteolytic processing of the viral polyproteins [14,15]. The papain-like proteases (PLpro) of SARS-CoV, MERS-CoV, HCoV-NL63 and SARS-CoV-2 have been reported to have deubiquitinating and deISGylating activities, which have been linked to the disruption of the host innate immune response [16,17,18,19,20]. On the other hand, the main proteases (Mpros), also known as 3C-like proteases (3CLpros), of several animal and human coronaviruses have also been reported to antagonize the host antiviral response by directly cleaving cellular proteins such as NF-kappa-B essential modulator (NEMO) [21,22,23,24] or signal transducer and activator of transcription 2 (STAT2) [25]. Furthermore, recent proteomics-based approaches have expanded the repertoire of possible cellular substrates of coronaviruses proteases [26,27,28], but additional work is needed to validate such substrates and to understand the biological relevance of such cleavage events in the context of virus infection. Also, in comparison with more highly pathogenic coronaviruses, the ability of HCoV-229E proteases (as well as that of other common cold HCoVs) to hijack antiviral host cellular responses is less well studied. The identification of conserved cellular proteins and pathways targeted by coronavirus proteases could help the development of antiviral strategies.

Recently, several studies have identified GSDMD as a substrate of viral proteases, including the 3Cprotease of enterovirus 71 (EV71) [29], the pS273R protease of the African swine fever virus (ASFV) [30], as well as the Mpros of highly pathogenic coronaviruses porcine epidemic diarrhoea virus (PEDV) and SARS-CoV-2 [26,31,32]. GSDMD is a cytoplasmatic protein, the function of which is to execute a lytic and proinflammatory form of cell death termed pyroptosis [33]. Upon recognition of PAMPs and DAMPs by PRRs, inflammasomes are assembled, and the latter recruit and activate inflammatory caspases (caspases-1 and -4/5), which in turn trigger the maturation of the pro-inflammatory cytokines interleukin-1β (IL-1β) and IL-18. Caspases-1 and -4/5 also activate GSDMD by releasing the autoinhibitory C-terminal domain of GSDMD from its N-terminal domain (GSDMD- N or p31), which then inserts into the cellular lipid bilayer and oligomerizes, forming pores that lead to cell swelling and release of cytokines [34]. Cleavage of GSDMD by PEDV and SARS-CoV has been proposed as a viral strategy to block or delay pyroptosis in virus-infected cells. However, other gasdermin family members can also exert pyroptotic activity in the context of virus infection. For example, GSDME has recently been shown to play a role in cell death induced by several viruses, including vesicular stomatitis virus (VSV), enterovirus 71 (EV71) and SARS-CoV-2 [32,35,36]. GSDME functions in a similar fashion to GSDMD. Under normal circumstances, GSDME remains inactive in the cytoplasm, but upon activation of the pro-apoptotic caspase-3, this can cleave GSDME after amino acid residue D270. The freed GSDME N-terminal fragment (GSDME-N or p30) can then migrate to the plasma membrane, where it oligomerizes and forms pores with pyroptotic activity [37]. Although recent studies have contributed to our knowledge on pyroptosis during virus infection and the interplay of gasdermins with viral proteins, cell death processes have been poorly characterized for common cold HCoVs such as HCoV-229E. Further studying such processes and the strategies these viruses use to circumvent them can provide new insights that may help understanding differences in pathogenicity between coronaviruses. Furthermore, the identification of common cell death processes among coronaviruses can pave the way for the development of pan-coronavirus antiviral strategies.

Here, we report that the main protease (Mpro) of HCoV-229E cleaves GSDMD at two sites within its N-terminal pore forming domain (Q29 and Q193), rendering the Mpro-generated GSDMD N-terminal fragments unable to induce lytic cell death. Despite GSDMD inactivation during virus infection, we showed that HCoV-229E-infected cells still undergo lytic cell death. We also found that GSDME is proteolytically activated by caspase-3 in the context of viral infection. Inhibition of caspase-3 in infected cells lead to reduced GSDME activation and concomitantly decreased cell death. In line with this, GSDME deficient cells displayed reduced lytic cell death during infection. These results suggest that GSDMD is inactivated by the virus Mpro and by cellular caspase-3 during HCoV-229E infection, but that infected cells still undergo pyroptosis partly mediated by GSDME.

## 2. Materials and Methods

### 2.1. Cell Culture and Virus Infection

HEK293T cells (ATCC, cat.no. CRL-3216) were cultured at 37 °C in 5% CO_2_ in Dulbecco’s modified Eagle’s medium (DMEM, Gibco, Waltham, MA USA) supplemented with 10% foetal calf serum (FCS, Bodinco BV, Alkmaar, The Netherlands), 100 units/mL penicillin, and 100 units/mL streptomycin (Sigma-Aldrich, Cat.no. P4458-100ML, St. Louis, MO, USA). Huh7 (a kind gift from Prof. Martin Schwemmle, Institute of Virology, University of Freiburg) and BEAS-2B cells were maintained in DMEM with 8% foetal calf serum, 100 units/mL penicillin, 2 mM L-glutamine (PAA Laboratories, Cölbe, Germany) and non-essential amino acids (NEAA, PAA Laboratories). H1299 cells (a kind gift from Marc Vooijs, Maastricht University) were maintained in DMEM supplemented with 8% foetal calf serum and 100 units/mL penicillin.

Infection with HCoV-229E (GenBank accession number NC_002645.1, [38]) was performed in DMEM containing 2% FCS, 2 mM L-glutamine, 100 units/mL penicillin, 100 units/mL streptomycin and NEAA at 33 °C in 5% CO_2_. All experiments with HCoV-229E were done in a biosafety level 2 laboratory.

### 2.2. Generation of BEAS-2B GSDMD and GSDME Knock out Cells

To produce lentiviruses, HEK293T cells were transfected with packaging vectors pMDLg/pRRE and pRSV-REV, envelope protein-expressing vector pCMV-VSVG [39], and the transfer vectors pEF1a-CAS9-2A-Blasticidin (#52962, Addgene) or pU6-gRNA-PGK-Puro-2A-BFP encoding CRISPR-Cas9 guide RNAs (gRNAs) against GSDMD (KO1: CCACAAGCGTTCCACGAGCGAAG; KO2: CCCTCTGGCCTCTCCATGATGAG), GSDME (KO1: CCTTGGTGACATTCCCATCCTCC; KO2: AGGCAATGGTGGTGGCAGCTGGG), or a non-target negative control gRNA (Human Sanger Arrayed Whole Genome Lentiviral CRISPR library, Sigma-Aldrich) using polyethylenimine (PEI 25K, Polysciences Inc., Eppelheim, Germany). At 24 h post transfection, medium was replaced, and supernatants were harvested at 72 h post transfection, centrifuged (1000× *g* for 10 min), and stored at −80 °C. To generate BEAS-2B GSDMD and GSDME knock out (KO) cell lines, BEAS-2B cells grown in 6-wells plates were first transduced with lentiviruses encoding Cas9 diluted in infection medium (DMEM, 2% FCS) containing 8 μg/mL Polybrene (Sigma Aldrich). At 16 h post transduction, medium was removed and replaced with culture medium. Three days post transduction, cells were passaged in selection medium containing 10 µg/mL blasticidin (ant-bl-1, InvivoGen, San Diego, CA, USA). BEAS-2B-Cas9 were later transduced with lentiviruses encoding for GSDMD, GSDME or non-target negative control gRNAs following the same procedure described above. Cells transduced with lentiviruses encoding for the different gRNAs were passaged in selection medium containing 10 µg/mL blasticidin and 5 µg/mL puromycin (ant-pr-1, InvivoGen) to obtain polyclonal cell populations lacking either GSDMD or GSDME. To evaluate the expression of Cas9 and of GSDMD or GSDME, protein lysates were obtained by harvesting cells in 100 μL 2xLSB and resolved by western blotting as described below.

### 2.3. Plaque Assay

Huh7 (2 × 10^5^ cells/well) or BEAS-2B (1.6 × 10^5^ cells/well) cells were seeded in 1 mL cell culture medium (DMEM, 8% FCS) in 12-well plates and incubated overnight at 37 °C in 5% CO_2_. The day after, 10-fold serial dilutions of supernatant samples were prepared in infection medium (DMEM, 2% FCS, 100 units/mL penicillin, 100 units/mL streptomycin, 2 mM L-glutamine and NEAA). Cell culture medium was removed, and 200 μL of serial dilutions were added to the wells. Cells were incubated at 37 °C with gentle rocking for 1 h, after which the virus inoculum was aspirated and 1 mL/well overlay medium (1.2% Avicel, 1% antibiotics, 2% FCS, and 50 mM HEPES in DMEM) was added. Cells were incubated at 33 °C for 4 days, when overlay medium was removed and cells were fixed with 1 mL/well 3.7% formaldehyde. After fixation, cells were stained 1.25% *w*/*v* crystal violet, and plaques were manually counted to determine the supernatant sample’s infectious virus titre.

### 2.4. Plasmids Used for Cell Culture Work and Transfections

Codon optimized sequences encoding for HCoV-229E-Mpro (amino acids 2954-3267 of pp1ab; NCBI Reference Sequence: NP_073549.1), HCoV-OC43-Mpro (amino acids 3236-3549; NCBI Reference Sequence: YP_009555238.1), and MERS-CoV-Mpro (amino acids 3242-3553 of pp1ab; NCBI ID: JX869059) with removed potential splice sites were cloned into a pcDNA3.1 expression vector (V79020, ThermoFisher, Waltham, MA, USA) in frame with a C-terminal V5-tag. Q3267 in the pp1ab of HCoV-229E Mpro, Q3549 in the pp1ab of HCoV-OC43 and Q3553 in the pp1ab in MERS-CoV were changed to a proline (P) to prevent cleavage of the tag by the Mpro. Codon optimized sequences encoding for SARS-CoV-Mpro (pp1a amino acids 3241-3546) and SARS-CoV-2-Mpro (amino acids 3264-3569 of pp1ab, GenBank accession number: MN908947.3) were cloned into a pcDNA3.1 expression vector (V79020, ThermoFisher) in frame with an N-terminal V5-tag. Each pcDNA3.1(-) plasmid coding for each coronavirus protease was used as template for site-directed mutagenesis using the QuickChangeTM strategy to mutate the corresponding active site cysteine (C) to an alanine (A): C144A for HCoV-229E; C145A for HCoV-OC43, SARS-CoV and SARS-CoV-2; and C148A for MERS-CoV. The human ORF sequence encoding for GSDMD was obtained from the Viral Vector Facility (LUMC) human ORF (cDNA) library from Sigma-Aldrich. The gene ORF was obtained in pDONR223 entry vectors and amplified by PCR with overhang primers to introduce partial sequences of an N-terminal FLAG-tag and C-terminal Myc-tag. PCR products were then cloned into pCR2.1 TOPO vector using the TOPO TA Cloning Kit (Invitrogen Thermo Scientific). From the pCR2.1 TOPO vector, sequences were further amplified by PCR with overhang primers to complete the FLAG-tag and Myc-tag sequences, and cloned into pCR8 TOPO vector using the pCR8 GW TOPO TA Cloning Kit (Invitrogen Thermo Scientific). Next, the sequence was cloned into the mammalian expression vector pDEST-cDNA3 using Gateway LR Clonase II Enzyme mix (Invitrogen Thermo Scientific). The nucleotide sequences coding for GSDMD fragments 1-193, 1-275 (p31), 30-193, 30-275, 30-STOP, 194-STOP and 275-STOP were amplified from pDONR223-GSDMD and inserted into pcDNA3.1(-) using HindIII and XbaI. pDEST-cDNA3-FLAG-GSDMD-Myc and pcDNA3.1(-)-GSDMD 1-275 were used as templates to for site-directed mutagenesis using the QuickChangeTM strategy to substitute glutamine (Q) 29 and 193 for alanine.

Primers used for plasmid construction and site-directed mutagenesis were designed in Geneious version 10.2.6 (Biomatters, Auckland, New Zealand). All constructs were verified by Sanger sequence analysis.

### 2.5. Immunoblotting Analysis

Cells were harvested in ice-cold PBS, centrifuged for 5 min at 2000 rpm and lysed in 2xLSB (250mM tris-base pH 6.8, 4% SDS, 20% glycerol, 10 mM DTT, and 0.01% bromophenol blue). Protein lysates were subjected to SDS-PAGE using 12.5% SDS-PAGE gels, and then blotted onto either nitrocellulose 0.2 µm (GE10600004, Amersham) or polyvinylidene difluoride 0.2 µm (GE10600022, Amersham) membranes, followed by incubation with the corresponding primary antibodies. Primary antibodies used were mouse monoclonal against β-actin at 1:10,000 (A2228, Sigma-Aldrich), mouse polyclonal against V5 tag at 1:2500 (A85280-10, Antibodies.com, Cambridge, United Kingdom), rabbit polyclonal against FLAG tag at 1:500 (F7425, Millipore), rabbit polyclonal against Myc tag at 1:2000 (ab9106, Abcam, Cambridge, United Kingdom), rabbit polyclonal serum against HCoV-229E nucleocapsid protein at 1:5000 raised in-house, rabbit monoclonal against GSDMD N-terminal at 1:1000 (HPA044487, Sigma-Aldrich), rabbit polyclonal against GSDME at 1:1000 (ab215191, Abcam). Mouse monoclonal to β-actin, mouse polyclonal to V5 tag, rabbit polyclonal to FLAG tag and rabbit polyclonal to Myc tag antibodies were diluted in 0.5% casein in PBS containing 0.05% Tween-20 (PBST). Rabbit polyclonal serum to HCoV-229E nucleocapsid protein, and rabbit monoclonal against GSDMD N-terminal and rabbit polyclonal against GSDME were diluted in 5% dry fat milk in PBST. β-actin was used as loading control. Biotin-goat-α-mouse (31800, Invitrogen) or donkey-α-rabbit (A16033, Invitrogen), and Cy3-conjugated mouse-α-biotin #200-162-211 (200-162-211, Jackson Immuno Research) were used for fluorescent detection with an Alliance Q9 Advanced (UVITEC, Cambridge, UK) scanner, and IRDye^®^ 680RD goat α-mouse and IRDye^®^ 800CW goat α-rabbit were used for fluorescence detection with an Odyssey scanner (Licor, Lincoln, NE, USA).

### 2.6. Indirect Fluorescent Antibody Assay

Huh7 and BEAS-2B were grown on glass coverslips and infected with HCoV-229E at MOIs of 0.01, 1 and 3, or mock infected. At 8, 24, 48 and 72 hpi, cells were fixed with 3% paraformaldehyde (PFA) in phosphate-buffered saline (PBS). Coverslips were then washed three times with PBS and permeabilized by incubating in 0.2% Triton X-100 in PBS for 10 min at room temperature. After permeabilization, cells were washed three times with PBS for 10 min. Cells were incubated with a rabbit anti-HCoV-229E nucleocapsid serum (1:3000) generated in house in 5% FCS in PBS for 1 h at room temperature. Next, coverslips were washed three times for 10 min in PBS and subsequently incubated with the secondary antibody goat anti-rabbit alexa-488 antibody (Thermo Fisher/Invitrogen) (1:300). Simultaneously, nuclei were stained with Hoechst 33258 (ThermoFisher) (1:100) in 5% FCS in PBS in the dark for 1 h at room temperature. Cells were washed three times with PBS for 10 min and shortly dipped in miliQ before being embedded on glass slides with ProLong glass antifade mountant (P36984, ThermoFisher). Coverslips were imaged using a Leica DM6B fluorescence microscope and analysed with the Leica Application Suite X software (version 3.8.1.26810, Leica Microsystems, Wetzlar, Germany).

### 2.7. CytoTox96^®^ Assay

Lytic cell death was indirectly studied by measuring lactate dehydrogenase (LDH) release in cell culture medium using the CytoTox 96 nonradioactive cytotoxicity assay kit (G1780, Promega, Leiden, The Netherlands) according to the manufacturer’s instructions. Absorbance at 490nm was measured with an EnVision multiplate reader (PerkinElmer, Waltham, MA, USA).

### 2.8. Caspases Activation Assays

The activity of caspases-1, -3 and -8 was studied in Huh7 and BEAS-2B cells at 24 and 48 h post-treatment or post-infection using the Caspase-Glo^®^ 1 Inflammasome Assay, the Caspase-Glo^®^ 3/7 Assay, and the Caspase-Glo^®^ 8 Assay (G9952, G8091, G8201, all three from Promega), respectively. Assays were performed according to the manufacturer instructions, and luminescence was measured in an EnVision multiplate reader (PerkinElmer).

### 2.9. Statistical Analysis

Data obtained by LDH release assay, plaque assay and caspase activation assay were analysed with GraphPad Prism 9.3.1. Data are presented as mean ± standard error of the mean (SEM). Unpaired Student’s *t* test or two-way ANOVA were used to assess statistical significance, and *p*-values of <0.05 were considered as statistically significant.

## 3. Results

### 3.1. Mpro of HCoV-229E Directly Cleaves GSDMD in an Overexpression System

To study whether Mpros of human coronaviruses of different pathogenicity target GSDMD, we first constructed mammalian expression plasmids encoding V5 tagged Mpro of MERS-CoV, SARS-CoV, SARS-CoV-2, HCoV-OC43 and HCoV-229E. Additionally, to assess whether the proteolytic activity of the above listed Mpros is essential for possible GSDMD proteolytic processing, we generated plasmids coding for the respective catalytic mutant Mpros by replacing the active site cysteine (C) residue by an alanine (A). Mpros exert their proteolytic activity by means of their catalytic dyad. This contains histidine (H) and cysteine as catalytic residues (Figure 1A), and mutations of the catalytic cysteine residue in coronaviruses Mpros are known to abrogate their proteolytic activity [40].

We then transfected HEK293T cells with a plasmid expressing N-terminal FLAG and C-terminal Myc tagged GSDMD in combination with plasmids encoding V5 tagged wild type (WT) or catalytic inactive mutant Mpro (C>A) of either MERS-CoV, SARS-CoV, SARS-CoV-2, HCoV-OC43 (Figure 1B) or HCoV-229E (Figure 1C). At 24 h after transfection, cells were lysed and protein lysates were examined by immunoblotting to detect Mpro-mediated GSDMD cleavage fragments. Western blot results show that the expression of WT Mpros decrease the abundance of full length (FL) GSDMD. In addition, in the presence of Mpro, we observed that bands of lower molecular weight than FL GSDMD appeared. The anti-FLAG blot showed a band of approximately 25 kDa corresponding to a GSDMD N-terminal fragment, whereas in the anti-Myc blot, we could observe a band of around 30 kDa that corresponds to a C-terminal fragment of GSDMD. Overexpression of FLAG-GSDMD-Myc together with the different catalytic mutant Mpros did not lead to the appearance of the 25 kDa and the 30 kDa fragments that are observed when GSDMD is co-expressed WT Mpros (Figure 1B,C). To further prove that Mpro’s proteolytic activity is essential for GSDMD processing, we co-expressed FLAG-GSDMD-Myc with HCoV-229E Mpro in the absence of or presence of increasing concentrations of the coronavirus Mpro inhibitor GC376 (0.5, 2.5 and 10 µM). GC376 was first identified as an inhibitor of the feline infectious peritonitis virus (FIPV) Mpro [41], but has been shown to possess inhibitory activity against the Mpros of several coronaviruses, including HCoV-229E [42]. Increasing concentrations of GC376 gradually rescued the expression of FL GSDMD. This observation is accompanied by a decreased intensity of the GSDMD C-terminal fragment when cells are treated with GC376 (Appendix A).

Although these data point to a direct cleavage of GSDMD by the Mpros of coronavirus of different pathogenicity, it has been well described that GSDMD is a substrate of three different caspases, the pro-inflammatory caspase-1 and the apoptotic caspases-3 and -8 (reviewed in [43]). To exclude that the GSDMD fragments we observed in the presence of WT Mpros are a result of GSDMD cleavage by any of the abovementioned caspases, we treated HEK293T cells co-transfected for FLAG-GSDMD-Myc and HCoV-229E WT Mpro expression with the pan-caspase inhibitor z-VAD. Treatment with z-VAD does not prevent cleavage of FL GSDMD in the presence of WT Mpro (Figure 1D). Altogether, these results show that Mpros of common cold coronaviruses HCoV-OC43 and HCoV-229E can cleave GSDMD and confirm previous data showing the same activity for the Mpros of highly pathogenic coronaviruses MERS-CoV, SARS-CoV and SARS-CoV-2.

### 3.2. HCoV-229E Mpro Cleaves GSDMD after Glutamines at Positions 29 and 193, and Resulting Fragments Lose Pyroptotic Activity

It has been recently reported that the Mpros of the pig coronaviruses PEDV and porcine deltacoronavirus (PDCoV), as well as the Mpros of the highly pathogenic coronavirus SARS-CoV, SARS-CoV-2 and MERS-CoV can cleave GSDMD at Q193 [31,32]. In addition to this cleavage site, a GSDMD peptide resulting from a cleavage event at position Q29 was reported in an N-terminomics study aiming to discover host cellular substrates of SARS-CoV-2 Mpro [26]. We then sought to test whether the cleaved GSDMD fragments observed in blots were a result of HCoV-229E Mpro cleavage within the described sites. To do so, we constructed two single GSDMD mutants in which the glutamines at positions 29 and 193 were replaced by an alanine (Q29A and Q193A, respectively), as well as a double mutant (DM) containing both amino acid substitutions (Figure 2A). Co-expression of the GSDMD Q29A mutant with HCoV-229E WT Mpro led to both a decrease in FL GSDMD and the appearance of the 25 kDa and 30 kDa fragments previously seen when overexpressing WT GSDMD with WT Mpro. However, when GSDMD Q193A was expressed together with WT Mpro, such fragments were no longer visible on the blot, while a new C-terminal fragment of approximately 50 kDa appeared. Finally, DM GSDMD expression in the presence of Mpro did not result in the degradation of the full-length protein, and no bands corresponding to Mpro-cleaved fragments could be identified by immunoblotting as shown in Figure 2B. The same results were obtained when the different GSDMD mutants were co-expressed with the Mpros of MERS-CoV, SARS-CoV, SARS-CoV-2 and HCoV-OC43 (Figure 2C). These results indicate that Mpro of HCoV-229E can cleave both after Q29 and Q193 within GSDMD.

Next, we asked what are the functional consequences of GSDMD cleavage by HCoV-229E Mpro. GSDMD consists of two domains, the active pore forming N-terminal 31 kDa (GSDMD-N or p31) and the C-terminal 22 kDa (GSDMD-C) domains (Figure 2A), which are separated by a linker region. Under normal conditions, FL GSDMD is inactive as GSDMD-C auto-inhibits GSDMD-N [33,44]. After interdomain cleavage at the linker region (at D275) by inflammatory caspase-1 or apoptotic caspase-8, GSDMD-N is freed and translocates to the plasma membrane where it oligomerizes to form pores that will eventually lead to cell swelling and membrane rupture, resulting in pyroptotic cell death [33]. As we have shown that HCoV-229E Mpro can cleave GSDMD within its N-terminal domain, we next aimed at studying whether the resulting Mpro-generated N-terminal fragments would retain pyroptotic activity. To do this, expression vectors were constructed to express the following GSDMD fragments: 1-193, 30-193 and 30-275 (Figure 2D). As a positive control for induction of pyroptosis, we built a plasmid to express the caspase-1 or -caspase-8-generated GSDMD p31 fragment (1-275). The different GSDMD fragments were expressed in HEK293T, and 24 h after transduction supernatants were collected to quantify LDH release as a proxy measurement for lytic cell death. As shown in Figure 2E, the expression of different Mpro (1-193, 30-193)- or Mpro and caspases-1/-8 (30-275)-generated N-terminal fragments did not lead to an increase in LDH release as compared to the empty vector (EV) condition, whereas the expression of GSDMD p31 fragment resulted in a significant increase in LDH release in the supernatant. These results suggest that Q29 and Q193 cleavages by Mpro within the N-terminal domain of GSDMD abolish its pyroptotic activity.

### 3.3. HCoV-229E Infection Causes Lytic Cell Death in BEAS-2B and Huh7 Cells

HCoV-229E can infect a variety of cell lines and induce cell death [45,46,47]. However, lytic cell death and the role of gasdermins in the context of HCoV-229E infection has been poorly studied. Therefore, to study whether HCoV-229E can cause lytic cell death, we first infected human bronco-epithelial BEAS-2B cells and human hepatocyte-derived carcinoma Huh7 cells, and measured LDH release at various time points post infection (24, 48 and 72 hpi). As shown in Figure 3A,B, HCoV-229E infection led to a significant increase in LDH release levels in an MOI and time-dependent manner in both BEAS-2B and Huh7 cells, respectively. Further, immunofluorescence microscopy revealed that HCoV-229E infection leads to morphological characteristics of cell death, including cell shrinkage and detachment (Figure 3C,D). Clear traits of cell death in virus-infected cells coincided in time with an increase in measured LDH release in the supernatants of the same cells. Taken together, these data suggested that HCoV-229E infection in both BEAS-2B and Huh7 cells induces cell death, and that at least a fraction of it is lytic.

### 3.4. HCoV-229E Infection Induces Cleavage of Both GSDMD and GSDME

As we observed lytic cell death in HCoV-229E-infected cells, we next aimed at studying the status of the pyroptotic executioner GSDMD in the context of virus infection. In addition, we also focused our attention onto gasdermin E, which has recently been shown to play a role in cell death induced by several viruses. To determine whether HCoV-229E infection has an effect on GSDMD and GSDME, BEAS-2B and Huh7 cells were mock-infected or infected with HCoV-229E. Cell lysates were collected and prepared for immunoblotting analysis at different time points after infection. As infection progressed, we could observe a slight decrease in band intensity for FL GSDMD and GSDME in whole cell lysates (WCL), but not a clear appearance of GSDMD and GSDME cleaved products (Appendix A). As GSDMD and GSDME mediate lytic cell death and we previously observed that virus-infected cells undergo lytic cell death and detach from the bottom of wells (Figure 3), we decided to look at GSDMD and GSDME in the supernatants of infected cells. Indeed, GSDMD and GSDME fragments were clearly visible in supernatants of infected cells, particularly at 72 hpi (Figure 4B,C). Using an anti-GSDMD N-terminal antibody, we were not able to detect the active p31 fragment. Instead, we observed the appearance of a band of approximately 40 kDa, most probably corresponding to the inactive p43 fragment resulting from cleavage by caspase-3/7 at position D88 within GSDMD (Figure 4B,C). Similarly, an antibody against the C-terminus of GSDMD also identified the p43 fragment, but not the p22 C-terminal counterpart fragment of active p31. For GSDME, a clear product of around 35 kDa appeared in the supernatant of infected cells as compared to mock cells, probably corresponding to the active p30 generated upon cleavage after D270 by caspase-3. These results suggest that HCoV-229E induces inactivation of GSDMD mediated by caspases-3/7 in host cells, whereas GSDME is activated, suggesting that GSMDE plays a role in virus-induced lytic cell death.

In previous overexpression experiments (Figure 1), we clearly showed that the Mpro of HCoV-229E can cleave GSDMD, which resulted in the appearance of N-terminal and C-terminal GSDMD fragments of around 25 kDa and 30 kDa, respectively. However, in the context of infection we did not identify such products by immunoblotting analysis. We then asked whether virus-expressed Mpro is able to cleave GSDMD in the context of infection. To address this, we expressed exogenous tagged GSDMD in human non-small cell lung carcinoma H1299 cells, and later infected them with HCoV-229E at an MOI of 1. By doing so, we hypothesized that if cleavage of GSDMD would occur during virus infection, resulting GSDMD fragments would be more easily detected by immunoblotting analysis. In addition, we also infected cells previously transfected with a plasmid for the expression of GSDMD Q29A+Q193A, in which Mpro cleavage sites were removed and therefore this molecule is not cleaved by Mpro. As shown in Figure 4D, immunoblotting analysis could identify a GSDMD N-terminal fragment of approximately 35 kDa at 48 hpi in cells transfected with FLAG-GSDMD WT-Myc. Such fragment was previously identified in co-expression experiments (Figure 1A), and was not detected in infected H1299 cells expressing the FLAG-GSDMD Q29A+Q193A-Myc. Although we have not been able to detect Mpro-mediated fragments of endogenous GSDMD in virus-infected cells, these results suggests that Mpro is capable of cleaving GSDMD in the context of virus infection.

### 3.5. Pan-Caspase Inhibition Dampens Virus-Induced Lytic Cell Death and Sustains Release of Infectious Virus Particles Overtime

Immunoblotting analysis of virus-infected cells pointed to the predominant activation of caspase-3 as revealed by the detection of GSDMD p43 and GSDME p30 fragments. Therefore, we next aimed at profiling the landscape of caspases activation in virus-infected cells. Besides quantifying the activity of caspases-3/7, we also assessed the activation of caspases-1 and -8 as both caspases have been described to proteolytically activate GSDMD and therefore participate in pyroptotic cell death [33,48]. In addition, it has been reported that caspase-8 is activated in SARS-CoV-2 infection of lung epithelial cells to trigger cell apoptosis and inflammatory cytokine processing [49]. Caspase activity assays revealed that all caspases tested show increased activity in virus-infected cells as compared to mock infected cells at 48 hpi, both in BEAS-2B (Figure 5A–C) and Huh7 cells (Appendix A), respectively. Interestingly, the increase in the activity of caspases during infection coincides in time with the increase in lytic cell observed earlier (Figure 3), suggesting that caspases may be partly responsible for virus-induced cell death. To follow up on this hypothesis, we next asked whether selective inhibition of caspases-1, -3 and -8, or inhibition of all caspases would decrease virus-induced cell death. Treatment of virus infected cells with Ac-YVAD-cmk (caspase-1 specific inhibitor) showed no decrease in LDH release as compared to vehicle (DMSO)-treated cells at 48 and 72 hpi. On the other hand, selective inhibition of caspases-3/7 with z-DEVD-fmk and of caspase-8 with z-IETD-fmk resulted in a reduction in LDH activity in the supernatant of virus-infected cells as compared to DMSO treated cells. Finally, HCoV-229E-infected cells treated with the pan-caspase inhibitor z-VAD also showed a significant decrease in LDH release at 72 hpi (Figure 5D and Appendix A). Additionally, we showed that selective inhibition of caspases-3/7 and -8, as well as pan-caspase inhibition, resulted in a reduction in or disappearance of GSDMD p43 and GSDME p30 fragments in the supernatant of virus-infected cells (Appendix A). Finally, we asked ourselves whether delaying and decreasing cell death with caspases inhibitors would have an effect in virus replication. To address this question, we quantified virus titres in the supernatants of infected cells treated with pan-caspase inhibitor z-VAD. Although infectious virus particle production starts between 4 and 8 h post-infection and peak virus titres in both BEAS-2B and Huh7 cells are obtained at around 16 to 24 h post-infection when cells are infected with a high MOI (Appendix A), we only measured virus titres at time points post-infection at which lytic cell death starts occurring. Figure 5E shows a significant increase in virus titres in the supernatants of z-VAD-treated BEAS-2B cells harvested at later time points during infection (72, 96 and 120 hpi). In Huh7 cells, we also observed an increase in the number of infectious virus particles in the supernatants of z-VAD treated cells harvested at 72 and 96 hpi as compared to vehicle-treated cells, but the difference was not statistically significant (Appendix A). Altogether, these data hint to an important role of caspases, particularly caspases-3 and -8, in the induction of lytic cell death during HcoV-229E infection, and to the fact that cell death limits sustained viral replication and the release of infectious viral particles.

### 3.6. GSDME Deficiency Reduces HCoV-229E Induced Lytic Cell Death in BEAS-2B Cells

Although our data indicate that caspases-3 and -8 are involved in lytic cell death upon HCoV-229E infection, it does not inform us on whether GSDMD and GSDME play a specific role in virus-induced cell death. To assess whether GSDMD and GSDME participate in the lytic cell death induced by HCoV-229E infection, we generated GSDMD and GSDME knock out (KO) BEAS-2B-Cas9 cells (Figure 6A). BEAS-2B-Cas9 transduced with a non-target gRNA were used as a control (Figure 6A). As shown in Figure 6B, a slight significant decrease in LDH release was observed for one of the GSDMD KO polyclonal cell lines (GSDMD KO#1) while the other GSDMD KO cell line (GSDMD KO#2) did not show a significant decrease as compared to control cells at 72 hpi. In contrast, a significant reduction in LDH release was observed in the two independent GSDME KO cell lines infected with HCoV-229E (Figure 6B). Further, we identified a human non-small cell lung carcinoma cell line (H1299) that does not express GSDMD (Figure 6C). Infection of H1299 cells with HCoV-229E led to lytic cell death in an MOI and time dependent fashion (Figure 6D), very similar to what we previously observed for BEAS-2B and Huh7 cells (Figure 3A,B). Additionally, GSDME was cleaved into its p30 active fragment in H1299 cells as shown by immunoblotting analysis (Figure 6E). These results together suggest that GSDME plays a more prominent role than GSDMD in mediating lytic cell death upon HCoV-229E infection.

### 3.7. HCoV229E Infection Leads to Increased Lytic Cell Death Levels in Cells Expressing GSDMD Q29A+Q193A

In previous experiments, we show that the Caspase-3/GSDME axis plays a more prominent role in virus-induced cell death in the context of HCoV-229E infection as compared to caspase-1/GSDMD axis. However, such observation could be explained by Mpro cleavage of GSDMD. Therefore, we next hypothesized that cells expressing a GSDMD mutant that cannot be targeted for cleavage by Mpro would be more susceptible to GSDMD-mediated lytic cell death. To this end, we transduced H1299 cells with lentiviruses in order to knock in either GSDMD WT or GSDMD Q29A+Q193A (Figure 7A). Knock-in cells were then infected with HCoV-229E at different MOIs and LDH release was measured at 48 and 72 hpi. At 48 hpi, no differences in LDH release levels were observed between H1299 GSDMD WT and H1299 GSDMD Q29A+Q193A, but a significant increase in LDH release was measured at 72hpi in H1299 GSDMD Q29A+Q193A cells as compared to H1229 GSDMD WT for the different MOIs tested (Figure 7B). These results suggest that GSDMD is cleaved by HCoV-229E Mpro during infection to dampen lytic cell death.

## 4. Discussion

An increasing number of studies suggest that human coronaviruses can utilize proteases encoded in their genomes to manipulate a variety of cellular processes in order to favour virus replication. Notably, the ability of both PLpros and Mpros of highly pathogenic coronaviruses SARS-CoV and SARS-CoV-2 to subdue antiviral immunity has been well documented [14,15], but the interaction of viral proteases from less pathogenic HCoVs with host proteins has drawn limited attention. Here, we report that the Mpro of HCoV-229E can proteolytically process the pyroptosis executioner protein GSDMD at two different sites within its N-terminal domain. Such cleavage events result in GSDMD fragments that lose the capability to induce lytic cell death, similarly to what has been recently reported for other coronaviruses [31,32], as well as for enterovirus 71 (EV71) and African swine fever virus (ASFV) [29,30]. By further investigating lytic cell death in the context of HCoV-229E, we also provide evidence showing that the caspase-3/GSDME axis is active during HCoV-229E infection and contributes to virus-induced lytic cell death.

Host proteins can harbour amino acid sequences that closely resemble to the preferred cleavage motif of Mpro: [small amino acid]-X-[L/F/M]-Q↓[G/A/S]-X (where X is any amino acid, glutamine (Q) residue is in P1 position, and ↓ is the cleavage site) [26,50]. If in the context of infection Mpro is present in close proximity to such proteins, the viral protease could potentially cleave them, leading to loss of function or a change in the activity of that particular protein. In this study, we show that GSDMD can be cleaved by the Mpro of HCoV-229E as well as by that of other human coronaviruses (Figure 1A,B). Our findings are in line with a recent report in which GSDMD is identified as a substrate of SARS-CoV-2 and MERS-CoV Mpros [32]. Interestingly, the Mpros of two porcine coronaviruses, PEDV and porcine deltacoronavirus (PDCoV), have also been described to cleave both porcine and human GSDMD [31]. In addition to the previously reported cleavage at Q193, we here describe that Mpros of different human coronaviruses can also cleave GSDMD within its pore forming domain residue Q29, a putative SARS-CoV-2 Mpro cleavage site only identified previously in a liquid chromatography–mass spectrometry-based N-terminomics in vitro analysis [26], and probably also observed in a more recent study, although it went undiscussed in that paper [32]. As expected, and unlike GSDMD p31 fragment (aa 1 to 275), GSDMD fragments resulting from cleavage by Mpro are rendered uncapable of inducing pyroptosis (Figure 2D), a preliminary indication that coronaviruses could inhibit GSDMD-mediated pyroptosis by proteolytically targeting GSDMD. Interestingly, we could not detect Mpro cleaved fragments of endogenous GSDMD in virus infected cells, although we did observe a decrease in full length GSDMD during HCoV-229E infection (Appendix A). However, detection of endogenous GSDMD fragments produced by the proteolytic activity of viral proteases has proven challenging in previous studies as well [29,30,31,32]. Nonetheless, we were able to show that ectopically expressed GSDMD is cleaved in HCoV-229E infected cells (Figure 4D), further suggesting that HCoV-229E Mpro can cleave GSDMD during infection.

The different members of the gasdermin family have been linked to a variety of cellular processes, but most predominantly they have been associated with inflammatory responses and lytic cell death (reviewed in [51]). GSDMD has been shown to play a role in pyroptosis induced by several pathogens (reviewed in [52,53]), and its activation is dependent on the assembly of inflammasomes upon sensing of PAMPs or DAMPs by cytosolic sensor proteins. Recent work on coronaviruses has identified several inflammasomes that can be activated during virus infection. NLRP3 inflammasome, mainly expressed in myeloid cells, is activated upon infection by mouse hepatitis virus (MHV), SARS-CoV, MERS-CoV and SARS-CoV-2 [54,55,56,57,58,59]. Interestingly, it has been recently shown that NLRP1 and CARD8 inflammasomes can sense the activity of viral proteases for downstream activation of pro-inflammatory caspase-1 [32,60]. Cleavage within the disordered N-terminus of CARD8 is conserved for the Mpros of several human coronaviruses, including HCoV-229E, and can lead to the activation of the CARD8 inflammasome and subsequent release of pro-inflammatory cytokines [60]. Similarly, NLRP1 is also cleaved by the Mpros of SARS-CoV-2 and MERS-CoV, leading not only to its activation and the ulterior recruitment and activation caspase-1, but also to the activation of the caspase-8/caspase-3/GSDME axis [32]. Additionally, NLRP1 has been recently identified as a sensor of double-stranded RNA (dsRNA) [61,62], but further research is needed to elucidate whether dsRNA arising from coronavirus replication can activate NLRP1. Here, we show that caspase-1, caspase-3 and caspase-8 are active in HCoV-229E infected cells (Figure 5A–C), although we did not connect such events with inflammasome activation. In future studies, it would be interesting to further characterize what cellular pathways lead to activation of such caspases in the context of HCoV-229E infection. Nonetheless, by specifically inhibiting caspases during HCoV-229E infection, we show that caspases-3 and -8 play a role in virus-induced cell death, whereas this is not the case for caspase-1 (Figure 5D,E). These results align with the predominant observation of inactive GSDMD p43 and active GSDME p30 fragments in infected cells (Figure 4B,C).

Recent studies have identified GSDME as an executor of lytic cell death in the context of infection with different RNA virus. GSDME mediates pyroptosis in human primary keratinocytes infected with vesicular stomatitis virus (VSV) [35]. VSV infection induces translation inhibition, which results in the depletion and inactivation of translation shut-down sensors Mcl-1 and Bcl-xL, respectively. This in turn induces mitochondrial damage, caspase-3 activation and subsequent activation of GSDME, which leads to the release of LDH and proinflammatory cytokine IL-1α. In SARS-CoV-2 infected normal human bronco epithelial (NHBE) cells, depletion of NRLP1 leads to decreased LDH release [32]. In NLRP1 deficient cells, virus infection fails to induce caspase-8/-3 activation and subsequent GSDME cleavage. Accordingly, knocking out GSDME in NHBE cells results in less LDH release upon virus infection. GSDME has also been shown to significantly contribute to lytic cell death upon influenza A virus (IAV) infection [62,63], EV71 infection [36], and Zika virus infection [64]. In line with these recent findings, we show that both cells treated with caspase-3 specific inhibitor z-DEVD-fmk and GSDME KO cells release less LDH as compared to their respective control cells (Figure 5D,E and Figure 6B). Interestingly, although GSDME appears to play a prominent role in virus induced pyroptosis, its deletion does not fully abrogate lytic cell death, an observation also described for the highly pathogenic SARS-CoV-2 [32]. Results from recent studies indicate that virus infection can activate components of apoptosis, necroptosis and pyroptosis simultaneously, and suggest that an extensive crosstalk between different forms of programmed cell death takes place during infection, a phenomenon recently termed as PANoptosis [65,66]. This could explain the still significant cell death levels in GSDME deficient cells, and calls for further research to shed light on the different mechanisms that drive cell death during HCoV-229E infection.

We show here that cells reconstituted with GSDMD Q29A+Q193A elicit increased lytic cell death upon HCoV-229E infection as compared to cells reconstituted with GSDMD WT (Figure 7B). A similar observation was also reported in A549 cells expressing a SARS-CoV-2 Mpro uncleavable GSDMD mutant (GSDMD Q193A) [32]. These results do not only suggest that Mpro of coronaviruses can proteolytically inactivate GSDMD in the context of infection, but also indicate that coronavirus infection triggers cellular pathways that eventually result into the activation of GSDMD. In addition to this, evidence points to a predominant caspase-3/GSDME activation during infection with HCoV-229E, as well as with other coronaviruses and RNA viruses [32,36,45,62,63,64,66]. However, it is still unclear whether gasdermins D and E serve as a host antiviral strategy or whether, on the contrary, they have a proviral role. It has been shown that different forms of lytic cell death can curb viral replication. Recently, it has been suggested that GSDMD and GSDME are important to limit the generation of IAV infectious virus particles [62]. Additionally, mitochondrial-mediated pyroptosis restricts VSV replication in a human skin model [35], while NLRP1 and NLRP9b-dependent pyroptosis can limit the generation of SARS-CoV-2 infectious virus particles in epithelial cells and the replication of rotavirus in mouse intestine, respectively [32,67]. Although we did not identify an effect of GSDMD or GSDME in virus replication, we did observe an increase in the release of infectious virus particles at late time points during infection in BEAS-2B cells when cells were treated with pan-caspase inhibitor z-VAD (Figure 5A,B), which suggests that caspase-mediated cell death could be a mechanism to limit virus replication and spread. This is in line with a study showing that inhibition of caspases in macrophages infected with MHV-3 increased virus yields and enhanced virus spread to neighbouring cells [68]. Further research is needed to better understand the effects of different forms of cell death on coronavirus replication.

In summary, the Mpro of HCoV-229E can inactivate the pyroptosis executioner GSDMD by cleaving it at two different sites within its N-terminal pore-forming domain. Further, HCoV-229E leads to the activation of caspase-3, among other caspases, which further inactivates GSDMD, but at the same time activates GSDME. Our results suggest that the caspase-3/GSDME axis plays an important role in pyroptosis-induced by HCoV-229E, but additional research is needed to better characterize the different triggers of cell death and their role in virus replication. Finally, further delineating proinflammatory and lytic cell death processes that are shared among coronaviruses could provide insights for the development of pan-coronavirus antiviral strategies.

## Figures and Tables

**Figure 1 viruses-16-00898-f001:**
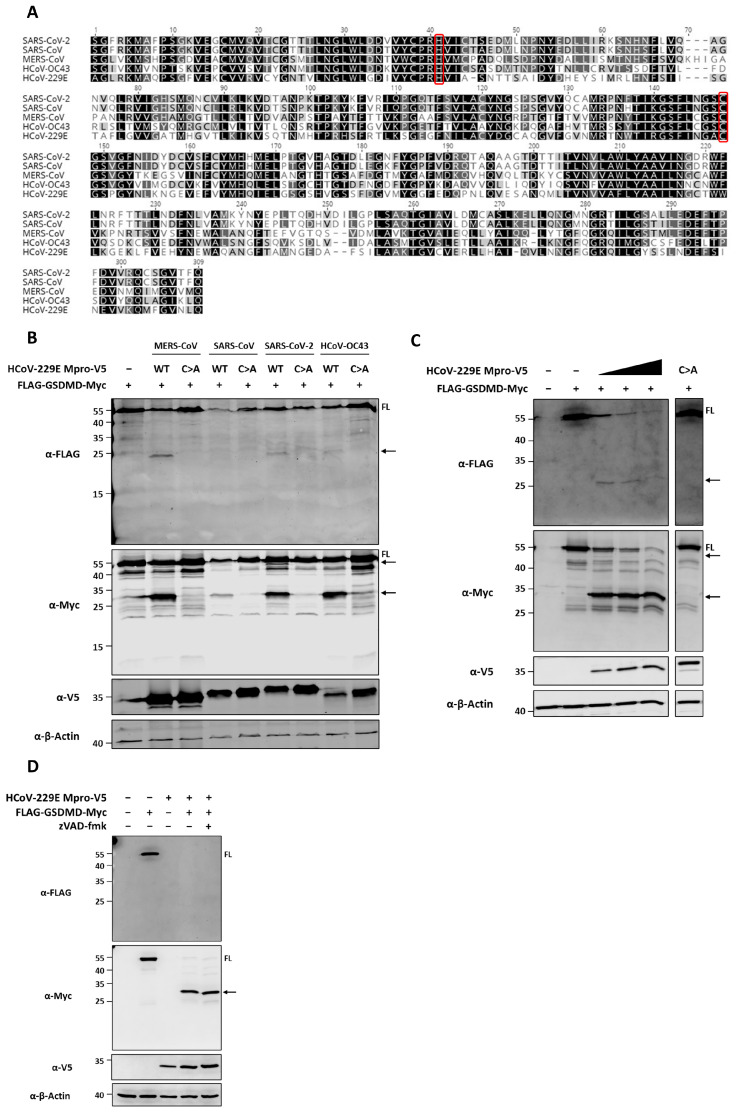
HCoV-229E Mpro cleaves GSDMD by means of its catalytic activity. (**A**) Amino acid sequence alignment for the Mpros of SARS-CoV-2, SARS-CoV, MERS-CoV, HCoV-OC43 and HCoV-229E in which the histidine and cysteine catalytic residues are circled in red; (**B**) HEK293T cells were transfected with a combination of a plasmid encoding FLAG-GSDMD-Myc (2 µg/well) and plasmids coding for V5-tagged MERS-CoV, SARS-CoV, SARS-CoV-2 or HCoV-OC43 WT Mpros or the respective catalytic mutant Mpros; (**C**) HEK293T cells were co-transfected with a plasmid encoding FLAG-GSDMD-Myc (2 µg/well) and increasing amounts of an HCoV-229E WT Mpro mammalian expression plasmid (0.5, 1 and 2 µg/well) or catalytic mutant Mpro (2 µg/well; and (**D**) HEK293T cells were left untreated or were treated with 25 µM zVAD for 2 h prior to transfection and were then co-transfected with the FLAG-GSDMD-Myc plasmid (2 µg/well) and V5-HCoV-229E Mpro plasmid. At 24 h post-transfection, cells were lysed for immunoblotting. The amino acid sequence alignment was generated with Geneious 10.2.6 and a similarity sequence scheme was applied using the Blosum62 score matrix with a threshold of 1 (black: 100% similarity; dark grey: 80–100% similarity; light grey: 60–80% similarity; white: <60% similarity). Immunoblots are representative of at least two independent experiments. FL, full-length; arrows indicate cleaved fragments identified by immunoblotting.

**Figure 2 viruses-16-00898-f002:**
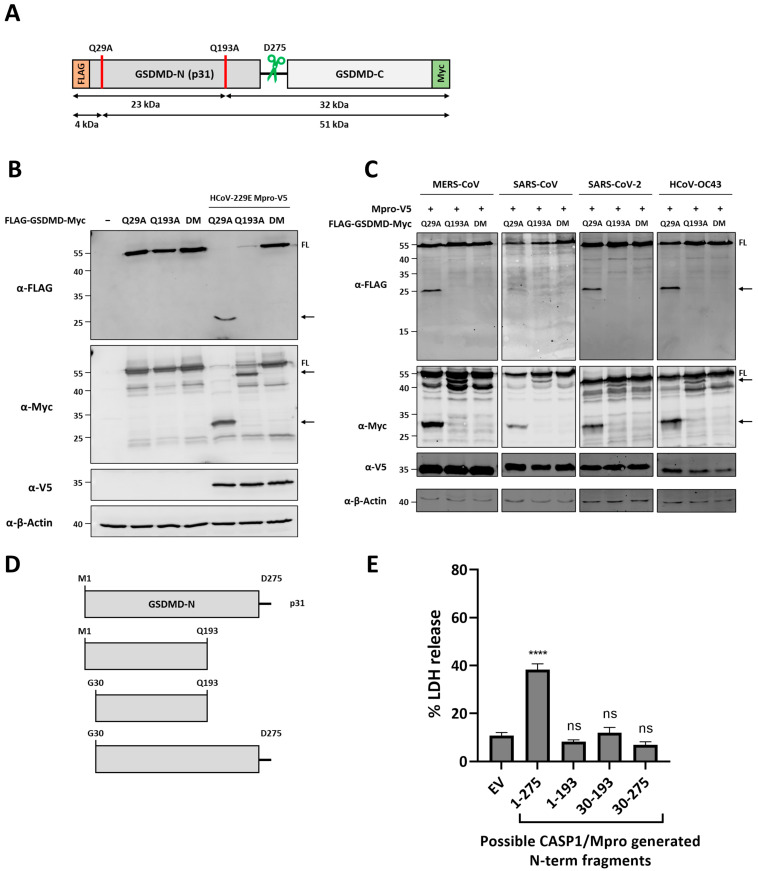
HCoV-229E Mpro cleaves GSDMD at Q29 and Q193, and resulting fragments lose pyroptotic activity. (**A**) Schematic diagram of FLAG-GSDMD-Myc structure in which the mutations at the two putative HCoV-229E Mpro cleavage sites (Q29A and Q193A, indicated as red lines) as well as the caspase-1 (green scissors) cleavage site are indicated; (**B**) HEK293T cells were transfected with plasmids coding for double tagged GSDMD Q29A, GSDMD Q193A or GSDMD double mutant (DM), or with a combination of a plasmids for the expression of the different GSDMD mutants and V5-HCoV-229E Mpro; (**C**) HEK293T cells were transfected with a combination of plasmids coding for double tagged GSDMD Q29A, GSDMD Q193A or GSDMD double mutant (DM), and WT Mpro of either MERS-CoV, SARS-CoV, SARS-CoV-2 or HCoV-OC43. At 24 h post-transfection, cells were lysed for immunoblotting. FL: full-length; arrows indicate cleaved fragments identified by immunoblotting; (**D**) schematic representation of GSDMD N-terminal fragments generated by HCoV-229E Mpro and/or caspase-1. Sequences coding for these fragments were cloned in mammalian expression vectors with no tags; and (**E**) plasmids coding for GSDMD N-terminal fragments 1-275, 1-193, 30-193 and 30-275 were transfected in HEK293T cells. An empty vector (EV) was used as a negative control. 24 h post-transfection, supernatants were harvested and used to quantify LDH release. Experiments were performed as triplicates and repeated at least thrice. FL, full-length; arrows indicate cleaved fragments identified by immunoblotting; LDH, lactate dehydrogenase; ns, not significant; ****, *p* < 0.00001.

**Figure 3 viruses-16-00898-f003:**
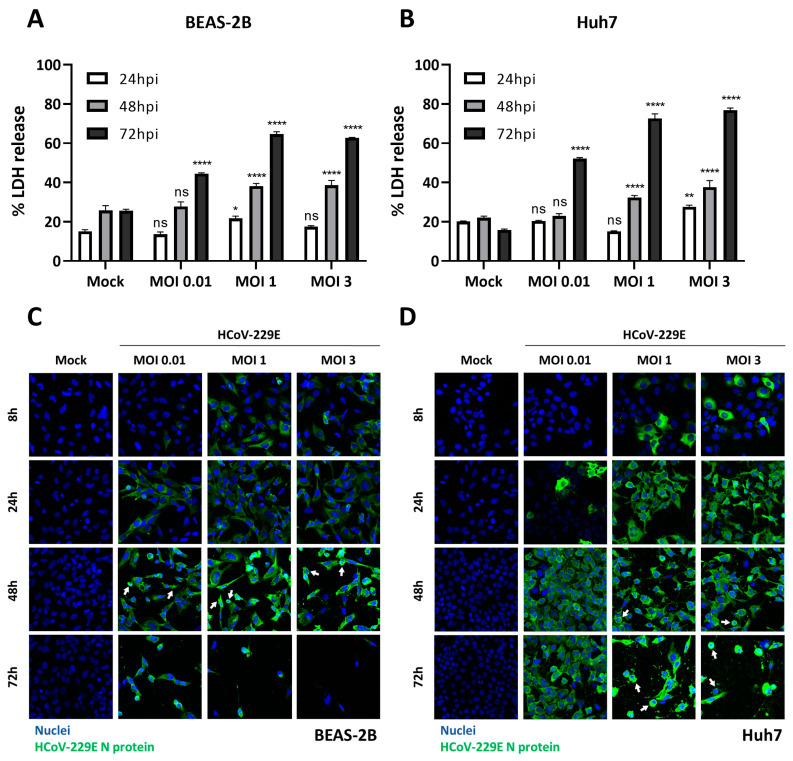
HCoV-229E induces lytic cell death. BEAS-2B (**A**,**C**) and Huh7 (**B**,**D**) cells were mock infected or infected with HCoV-229E at different MOIs as indicated. (**A**,**B**) At 24, 48 and 72 h post-infection, LDH release was assessed; (**C**,**D**) at the indicated time points after infection, cells were fixed and stained with an antibody against HCoV-229E nucleocapsid protein (green), and nuclei were counterstained with Hoechst staining (blue). White arrows indicate events of cell shrinkage. Experiments were performed as triplicates and repeated at least twice. LDH, lactate dehydrogenase; MOI, multiplicity of infection; ns, not significant; *, *p* < 0.01; **, *p* < 0.001; ****, *p* < 0.00001.

**Figure 4 viruses-16-00898-f004:**
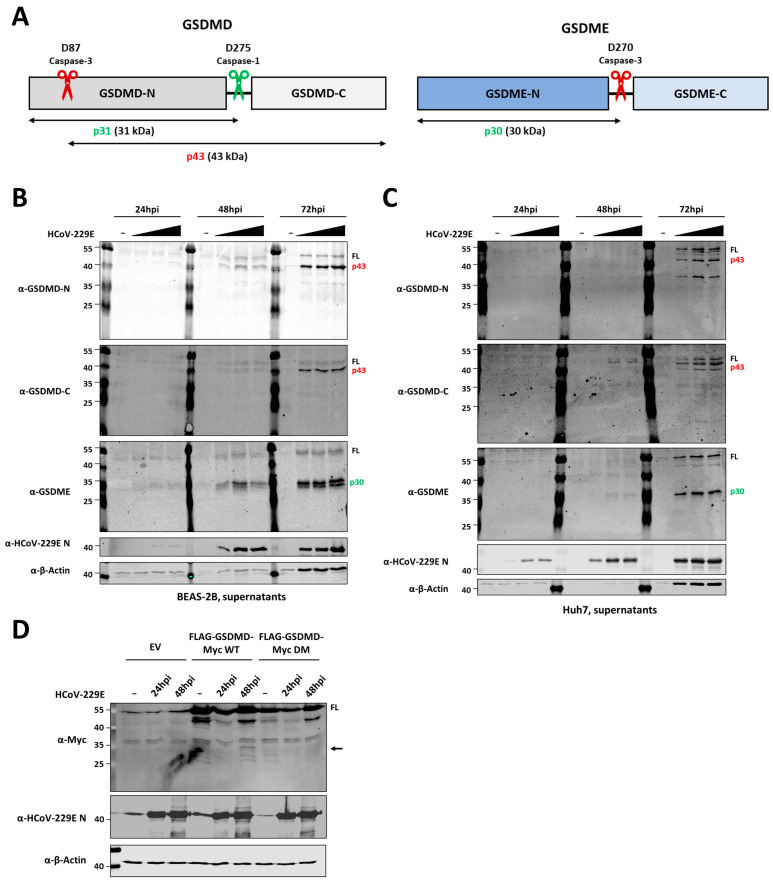
HCoV-229E infection leads to caspase-3 mediated cleavage of GSDMD and GSDME. (**A**) Schematic representation of GSDMD and GSDME proteins in which cleavage sites by caspases-1 (green scissors) and -3 (red scissors) and resulting GSDMD fragments are indicated; BEAS-2B (**B**) and Huh7 (**C**) cells were infected with HCoV-229E at MOIs of 0.01, 1 and 3. Supernatants were harvested at the indicated time points and analysed by immunoblotting for GSDMD N-terminal, GSDMD C-terminal, GSDME, HCoV-229E nucleocapsid protein and β-actin; and (**D**) H1299 cells were transfected with mammalian expression vectors for the expression of FLAG-GSDMD-Myc WT or FALG-GSDMD-Myc containing Q29A and Q193A mutations (double mutant, DM), and infected with HCoV-229E at an MOI of 1 at 24 h post-transfection. At the indicated time points, cell lysates were harvested and immunoblotted for Myc, HCoV-229E nucleocapsid protein and β-actin. Immunoblots are representative of at least two independent experiments. FL, full-length; arrows indicate cleaved fragments identified by immunoblotting.

**Figure 5 viruses-16-00898-f005:**
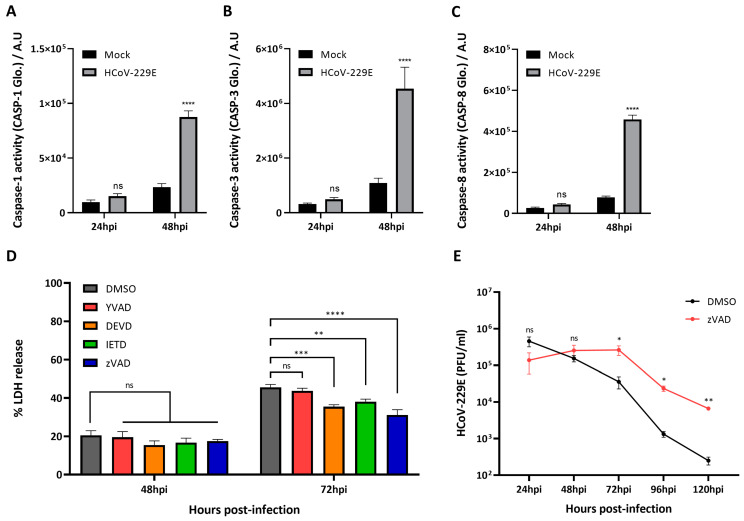
Pan-caspase inhibition dampens virus-induced lytic cell death and sustains release of virus infectious particles overtime. (**A**–**C**) BEAS-2B cells were mock infected or infected with HCoV-229E MOI 3 and caspase activity was measured at 24 and 48 hpi; (**D**) BEAS-2B and Huh7 cells were treated with vehicle (DMSO), 40 µM Ac-YVAD-cmk, 20 µM z-DEVD-fmk, 25 µM z-IETD-fmk or 25 µM z-VAD, and mock infected or infected with HCoV-229E at an MOI of 3, and LDH release was then assessed at the indicated time points; and (**E**) supernatants of infected BEAS-2B treated with either vehicle (DMSO) or z-VAD were harvested at the indicated time points, and HCoV-229E infectious virus particles were quantified by plaque assay. Experiments were performed as triplicates and repeated at least thrice. A.U, Arbitrary units; LDH, lactate dehydrogenase; MOI, multiplicity of infection; ns, not significant; *, *p* < 0.01; **, *p* < 0.001; ***, *p* < 0.0001; ****, *p* < 0.00001.

**Figure 6 viruses-16-00898-f006:**
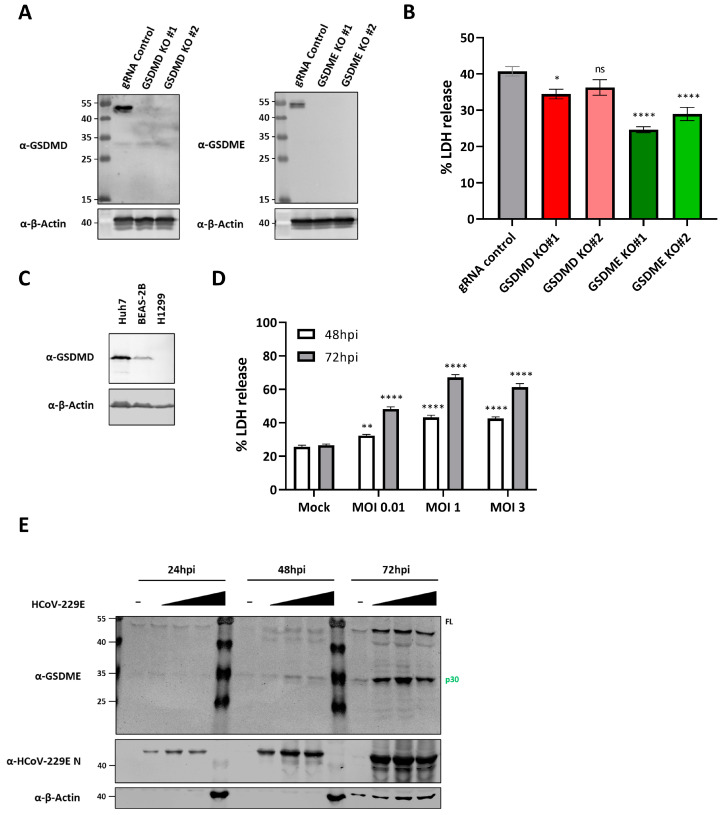
GDMDE contributes to HCoV-229E-induced lytic cell death. (**A**) BEAS-2B-Cas9 cells were transduced with lentiviruses encoding a non-target negative control gRNA, or with GSDMD or GSDME gRNAs. Cells transduced with lentiviruses encoding for the different gRNAs were passaged in selection medium containing 10 µg/mL blasticidin and 5 µg/mL puromycin to obtain polyclonal cell populations lacking either GSDMD (left) or GSDME (right). To confirm the knockout of GSDMD or GSDME, protein lysates were obtained by harvesting cells in 100 μL 2xLSB and resolved by immunoblotting; (**B**) the different BEAS-2B cell lines were mock infected or infected with HCoV-229E at an MOI of 3, and LDH release was then assessed at 72hpi; (**C**) protein lysates of Huh7, BEAS-2B and H1299 cells were harvested and analysed by immunoblotting for GSDMD and β-actin; (**D**) H1299 cells were mock infected or infected with HCoV-229E at MOIs of 0.01, 1 and 3, and LDH release was measured at the indicated time points; and (**E**) supernatants of infected H1299 cells were harvested at the indicated time points and analysed by immunoblotting for GSDME, HCoV-229E nucleocapsid protein and β-actin. Experiments were performed as triplicates and repeated at least thrice. LDH, lactate dehydrogenase; ns, not significant; *, *p* < 0.01; **, *p* < 0.001; ****, *p* < 0.00001.

**Figure 7 viruses-16-00898-f007:**
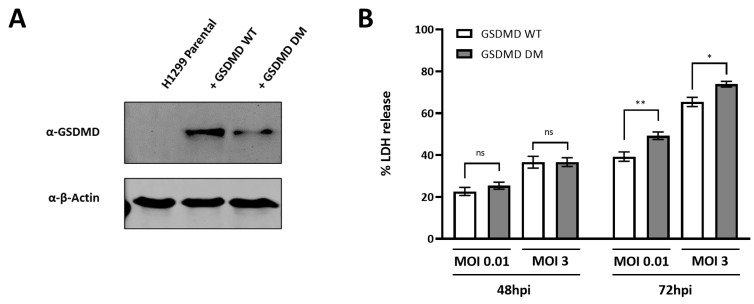
HCoV-229E infection leads to increased lytic cell death levels in cells expressing GSDMD Q29A+Q193A. (**A**) H1299 cells were transduced with lentiviruses encoding WT GSDMD or GSDMD Q29A+Q193A (double mutant, DM) and passaged in selection medium 2.5 µg/mL blasticidin to obtain polyclonal cell populations expressing either GSDMD WT or GSDMD DM. To confirm the knock-in of GSDMD WT or GSDMD DM, protein lysates were harvested and resolved by immunoblotting; and (**B**) H1299 GSDMD WT or H1299 GSDMD DM were mock infected or infected with HCoV-299E at MOIs of 0.01 or 3, and LDH release was measured at the indicated time points. Experiments were performed as triplicates and repeated at thrice. LDH, lactate dehydrogenase; MOI, multiplicity of infection; ns, not significant; *, *p* < 0.01; **, *p* < 0.001.

## Data Availability

All data supporting the reported results can be found in the published paper.

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
