# Peer review of "Human Coronavirus 229E Infection Inactivates Pyroptosis Executioner Gasdermin D but Ultimately Leads to Lytic Cell Death Partly Mediated by Gasdermin E"

_viruses, 2024, doi:10.3390/v16060898_

Round 1

Reviewer 1 Report

Comments and Suggestions for Authors

The authors have investigated the interplay of HCoV-229E with pyroptosis. Virus modulation of cell death is an area of intense research at this time as we learn more about the interconnected cell death pathways that are a component of our response to infection.

The work described here is important as it adds to our understanding of the many ways viruses interact with pyroptosis and its relevance to pathogenecity.

However, there are parts of the manuscript that require modification before this manuscript can be published.

1. In several places, the authors refer to conservation of cellular pathways targeted by proteases from pathogenic and non-pathogenic coronaviruses. But nowhere in the manuscript is any data presented where such a comparison is made, making these statements misleading.

Please perform such experiments and report the data or remove all such references, e.g. in Abstract, Introduction, results and discussion. This manuscript is about the interaction of HCoV-229E with pyroptotic pathways.

2. The authors should perform experiments where GSDMD and GSDME related pyroptosis is induced (e.g. chemically) either prior to or during infection. Such inductions should also be performed in the transfected cells that lack either GSDMD or GSDME. This would provide strong evidence whether HCoV-229E indeed inhibits pyroptosis.

3. The authors have used LDH release to study lytic cell death and have ascribed it to pyroptosis. How do they know it is not necrosis or necroptosis or late stage apoptosis? 

4. Given the high infection % shown in the manuscript, the authors did not detect cleaved GSDME in infected cells without ectopic GSDMD. Please discuss why this may have occurred.

5. Have the authors considered the possibility that late lysis, as shown in the manuscript, may aid virus release? Please discuss this aspect.

6. Please focus your discussion on specific, relevant topics. Currently it is a wide ranging, very long discussion often on aspects not investigated in the manuscript.

7. Please check all figure legends and ensure that the description matches the data shown in the figure.

Reviewer 2 Report

Comments and Suggestions for Authors

Human coronaviruses (CoVs) have a wide range of virulence, causing from common cold to severe respiratory infection that can be lethal. Knowledge of the virus-host interaction and the underlying molecular mechanisms is crucial to develop novel antiviral strategies. This manuscript describes the role of CoV main protease (Mpro) in the modulation cell death by cleavage of gasdermins GSDMD and GSDME. The manuscript findings have limited novelty. But it is worth noting that it focused in HCoV-229E, which is understudied in comparison with highly pathogenic CoVs; and in the absence of a good animal model, it uses several human cell lines including cells from the respiratory tract to do the analyses. The manuscript is well-written, the rationale is clear, and the experiments are justified. However, there are a few issues than could be considered to improve the manuscript.

Specific comments:

1. Fig. 5 and Supplementary figure 3. There are some issues requiring further clarification:

(i) Line 483. Please note that at 48 hpi, the difference between non-treated and treated cells is not significant for any condition analyzed

(ii) Line 490, “small but significant increase”. Please note that at 72 hpi the difference is almost 10-fold and at 120 hpi is more than 10-fold. Maybe this should not be considered a “small increase”

(iii) Any hypothesis to explain why zVAD inhibition increases viral titers in BEAS-2B cells but not in Huh-7 cells?

(iv) Since peak titers in Figs. 5E and Supplementary Fig. 3G are at 24 hpi, the graphs do not clearly demonstrate infectious virus production. Authors would consider to infect cells at a lower MOI or to also analyze earlier times post-infection

2. Fig. 1B and lines 282-284. Please note that, in the case of SARS-CoV, it seems there is a decrease of the full-length tagged GSDMD band. Partially related with this, why FLAG band was not detected after overexpression of SARS-CoV and HCoV-OC43 Mpros? Please, clarify these issues.

3. A point for discussion: As indicated by the authors, pyroptosis may have a correlation with cytokines release and inflammation. It would be nice to compare in gasdermins KO cells the cytokine release after mild vs. virulent human CoV infection.

Minor comments:

1. In general, HCoV-229E cell culture adaptation implies evolution of the virus and split of accessory gene ORF4 into ORF4a and ORF4b (such as in the virus used in this manuscript). Would this issue have any implications in the effects observed in this work after infection?

2. Fig. 4. Using the same color for caspase-1 cleavage in GSDMS and caspase-3 cleavage in GSDME could be confusing for the reader. Either use same color for same caspase in both proteins, or a new color for GSDME cleavage.

3. Figs. 3C and 3D. Please note that time points are not indicated.

4. Lines 37, 38, 49 and 50. Please note that taxonomic terms should be in italics.

5. Line 383. “infection can infect” could be replaced by “virus can infect”
